# Assessment, Treatment, and Follow-Up of Phlebitis Related to Peripheral Venous Catheterisation: A Delphi Study in Spain

**DOI:** 10.3390/healthcare12030378

**Published:** 2024-02-01

**Authors:** Alba Torné-Ruiz, Mercedes Reguant, Montserrat Sanromà-Ortiz, Marta Piriz, Judith Roca, Judith García-Expósito

**Affiliations:** 1Department of Nursing and Physiotherapy, University of Lleida, 25199 Lleida, Spain; alba.torne@udl.cat (A.T.-R.); montserratso@blanquerna.url.edu (M.S.-O.); judithga1127@gmail.com (J.G.-E.); 2Hospital Fundació Althaia, Xarxa Assistencial Universitària de Manresa, 08243 Manresa, Spain; 3Department of Research Methods and Diagnosis in Education, University of Barcelona, 08035 Barcelona, Spain; mreguant@ub.edu; 4Blanquerna School of Health Science, Ramon Llull University, 08025 Barcelona, Spain; 5Infectious Diseases Division, Hospital de la Santa Creu i Sant Pau, 08041 Barcelona, Spain; mpirizm@santpau.cat; 6Institut de Recerca de l’Hospital de la Santa Creu i Sant Pau, 08041 Barcelona, Spain; 7Health Care Research Group (GRECS), Biomedical Research Institute of Lleida, 25198 Lleida, Spain; 8Health Education, Nursing, Sustainability and Innovation Research Group (GREISI), 25199 Lleida, Spain; 9Group Preving, 03003 Alicante, Spain

**Keywords:** Delphi technique, intravenous infusion, nursing, peripheral venous catheter, phlebitis

## Abstract

Background: Phlebitis related to peripheral venous catheters (PVCs) is a common complication in patients who require these devices and can have important consequences for the patients and the healthcare system. The management and control of the PVC-associated complications is related to nursing competency. The present study aims to determine, at the national level in Spain, the consensus on the assessment, treatment, and follow-up of PVC-related phlebitis and the importance of the actions taken. Method: A three-round Delphi technique was used with clinical care nurses who are experts in the field of in-hospital intravenous treatment in Spain. For this, an online questionnaire was developed with three open-ended questions on the dimensions of phlebitis assessment, treatment, and follow-up. For the statistical analysis of the results, frequencies and percentages were used to determine consensus, and the measures of central tendency (mean, standard deviation, and the coefficient of variation) were used to rank importance. The coefficient of variation was set as acceptable at ≤30%. Results: The final sample was 27 expert nurses. At the conclusion of round 3, actions were ranked according to their importance, with six items included in the PVC-related phlebitis assessment (symptomatology/observation, redness, the Maddox scale, induration, temperature, and pain), two in treatment (catheter removal, pentosan polysulphate sodium ointment + application of cold), and just one in follow-up (general monitoring + temperature control). Conclusions: There is a major disparity in relation to the PVC-related phlebitis assessment, treatment, and follow-up actions. More clinical studies are therefore needed to minimise the complications associated with the use of PVCs, given their impact on the quality of care and patient safety and their economic cost.

## 1. Introduction

Vascular access is an essential element of healthcare. According to Almirante [1], it is estimated that 70% of patients have a peripheral venous catheter (PVC) inserted during their hospital stay. More specifically, a 2022 Spanish nosocomial infection prevalence study called the EPINE study [2] found that 76.34% of patients had PVCs inserted. The use of these devices has many advantages, but can also involve certain complications, with one of the most frequent being phlebitis [3]. Its prevalence varies widely from 20% to 80% of patients undergoing intravenous therapy [4]. Phlebitis is associated with endothelial damage in the intima layer (leading to the formation of a thrombus) and the inflammation of the tunica media of the vein, causing oedema, infiltration, and possibly the rupture of the integrity of the wall [5]. This process results in a wide range of symptoms, generally of a local character [6]. Some studies [7] have related phlebitis in adult patients with the presence of a chronic illness, the duration of catheter embedment, and the infusion type. Once phlebitis has developed, an immediate removal of catheter has been associated with an improvement in clinical signs and symptoms [8]. As symptoms make it difficult to continue therapy and cause patient discomfort [6,9], nurses’ knowledge and the early recognition of risk factors for the occurrence of phlebitis can reduce its complications [10].

There are 71 phlebitis rating scales, but most have not been adequately validated [11]. According to the newly published updates of the Infusion Therapy Standards of Practice in 2021 by the Infusion Nurses Society (INS) [12], it is recommended to use a standardised scale or definition of phlebitis that is valid, reliable, and clinically feasible, as well as the same method of assessment within an organisation. 

No consensus exists as to the optimal management of PVC-related phlebitis at the clinical level, although several therapies have been proposed in the literature, including topical and systemic treatments [6,8,13,14,15,16]. The INS [12] recommends applying warm compresses, elevating the limb, providing analgesics, or carrying out pharmacological interventions such as the administration of anti-inflammatory agents. 

For a problem in clinical practice as important as PVC-related phlebitis, the scarcity of pertinent studies on treatments for this complication or on the wide variety of available products is striking, as shown by the current reviews of the literature [6,14,15]. These studies reveal three important issues that need to be tackled. First, as previously mentioned, the lack of consensus on the assessment of phlebitis using validated instruments and even on the signs and symptoms associated with its identification. Second, the variability in its treatment, although most involve a topical application. Third, follow-up actions that include patient control or monitoring activities in health care. This third and final issue has been explored the least. 

The uncertainty, the lack of clear guidelines, and the high variability in this topic justify an exploratory study based on a Delphi-type consensus method. The Delphi method possesses appropriate characteristics to tackle this topic, with a structured and iterative approach that aims to gather the knowledge of a panel of experts in PVC-related phlebitis. The main aim of the present study is to determine, among expert hospital care nurses at the national level in Spain, the consensus on the management of PVC-related phlebitis and the importance of the actions taken. As a secondary aim, the actions that are taken are considered in terms of three dimensions: (1) assessment; (2) treatment; and (3) follow-up.

## 2. Materials and Methods

### 2.1. Design

The Delphi technique was used for this study. This is a structured process to systematically collect expert judgements on a problem, process the information, and, using statistical resources, determine the general consensus of the group and rank the solutions to the problem [17,18]. It is a social research technique that seeks to obtain a reliable collective opinion from a group of experts [19]. The guidelines (Appendix A) recommended for Delphi studies (CREDES) [20] were followed.

### 2.2. Stages and Tools

In accordance with the model proposed by García Valdés et al. [21], the stages shown in Figure 1 were followed to carry out the study. 

### 2.3. Preparation Stage: Selection of Experts

For the selection of the experts, general aspects such as willingness to participate, commitment to the activity, availability of time, and communication skills were assessed [17]. The inclusion criteria were as follows:Clinical nursing with practical experience in the field of intravenous therapy. Expertise was established on the basis of at least 10 years of nursing experience and professional practice in a second- or third-level hospital (without differentiating between hospital units). Second-level hospitals offer general care services and some specialist services of generally lower complexity. Third-level hospitals offer a full range of specialist care. The hospitals could be public, fully private, or private state-contracted hospitals.Participation in national projects such as Phlebitis Zero. This Spanish national project promotes the use of standardised interventions to stop phlebitis (http://flebitiszero.com/app/ (accessed on 15 March 2022)).Currently working in a second- or third-level hospital in Spain.

The exclusion criteria included nursing professionals who were not presently working or who were active at other healthcare levels. In addition, in the selection of the experts, maximum representativeness and participation with respect to the various autonomous communities was sought at the Spanish national level. 

Contact with the experts was made through an invitation email. The experts were initially found by different means: (1) the literature review of the topic (corresponding author); (2) the review of hospital documents and protocols (reference authors); and (3) contact with the coordinators of the Phlebitis Zero project to disseminate the proposed study. A total of 47 professionals were invited to participate. The number initially invited took into consideration a possible loss of around 40% in accordance with data from other studies [22].

### 2.4. Preparation Stage: Preparation of Tool

The tool used was an ad hoc online questionnaire with 3 open-ended questions based on the dimensions of assessment, treatment, and follow-up of PVC-related phlebitis in adult hospitalized patients (excluding patients undergoing oncological treatments):ASSESSMENT: How do you assess the degree of phlebitis associated with an upper extremity venous catheter in a patient?TREATMENT: What treatment do you apply in the event of PVC-related phlebitis?FOLLOW-UP: What follow-up activities are established?

In the first round, basic data were also collected from the participants: age (numerical data), sex (male, female, non-binary), initial training (graduate or nursing diploma), advanced training (postgrad, Master’s, PhD, speciality, none), years of experience (numerical data), area of expertise (open answer), and Autonomous Community (Andalusia, Aragon, Balearic Islands, Canary Islands, Cantabria, Castilla-La Mancha, Castilla and León, Catalonia, Community of Madrid, Navarre, Valencian Community, Extremadura, Galicia, Basque Country, Principality of Asturias, Region of Murcia and La Rioja).

### 2.5. Consultation Stage and Data Collection

A total of 3 rounds of online consultations were carried out through the web platform Research Electronic Data Capture (REDCap) (https://www.project-redcap.org/ (accessed on 15 May 2022)), following the model proposed by Reguant-Alvarez and Torrado-Fonseca [17]. In the first round, contact with the expert was by email, with a message containing a presentation of the project and a voluntary request for participation. In each successive round, the questionnaire was returned to each expert with the group’s previous answers, accompanied by the statistical analysis, and feedback was requested based on the results obtained. A summary of each round is presented below:In round 1, participants were asked to answer the three open-ended questions on the three dimensions (assessment, treatment, and follow-up) with regard to PVC-related phlebitis.In round 2, the results of the first consultation (i.e., the responses and their frequency) were incorporated, and the same participants were asked to express their agreement or otherwise with each response item (with a YES or NO) and rate its importance using a 5-point Likert scale: None (0), Little (1), Neutral (2), Quite (3), and Very (4). All responses were included for evaluation.In round 3, the results of the second round were presented to the same group: the percentages (%) of ‘agreement’, and the mean (M) and standard deviation (SD) values of ‘importance’. The experts were then asked to prioritise the importance by ranking the response items of each dimension numerically from 1 (least important) to the maximum number possible (which varied according to the number of response items in each dimension). In other words, the highest score was given to the most important item. To analyse these data, the mean and SD of the inverted ranking values were calculated, and finally, the coefficient of variation (CV) was calculated.

The data were collected during the period from 1 May to December 2022.

### 2.6. Consensus Stage: Statistical Analysis

Given the high variation found in the literature [23] with regard to the statistical measures used in this type of analysis, especially in the health sciences field, the following is a summary of the elements that the research team defined in the study. The consensus opinion of the group was obtained in each iterative round from the general opinion obtained in the open questions of the first round, the % agreement, the median, M and SD of the importance ratings for each of the response items provided by each expert in the second round of consultation, and the M and SD of the ranking of importance and their CV in the third and final round. According to the literature [24,25], a 50% variability is considered an acceptable value. However, given the differing responses, the research team made the decision to use a CV value of 30% following the guidelines proposed by Reguant-Álvarez and Torrado-Fonseca [17]. A CV value of less than or equal to 30% indicates that the arithmetic mean is a representative of the dataset and, therefore, the dataset is ‘homogeneous’. In contrast, if the CV exceeds 30%, the mean will not be a representative of the dataset (and so will be ‘non-homogeneous’). 

### 2.7. Methodological Quality of the Study

Experts were initially consulted to ensure the methodological quality of the study. The proposed questions were previously assessed by three clinical experts (nurses with more than 20 years of experience and MA qualifications) to ensure that the concepts included in the study were clear and easy to understand. As a result of this consultation, some minor semantic modifications to the questions were made. Secondly, the selected sample consisted of a panel of experts who met the inclusion and exclusion criteria described in Section 2.3 and who were representatives of the area of knowledge to be analysed. Thirdly, 3 successive rounds of the questionnaire were held until consensus was reached by the panel of experts on the assessment, treatment, and follow-up actions of PVC-related phlebitis and their importance assessment, treatment, and follow-up of PVC-related phlebitis. For this reason, the criteria of trustworthiness are ensured by the following: (1) social validity (clear and real aim); (2) the adequacy of data (expert participants and the follow-up of suitable strategies to obtain the data); (3) the adequacy of interpretation (researchers who are experts in the methodology used); and (4) the adequate management of subjectivity (the joint treatment of the qualitative data by the research team) [26]. 

### 2.8. Ethical Considerations

The Delphi model followed in this study is considered low in terms of risk and intervention [27]. It comprises a study conducted by experts without the intervention or participation of patients with phlebitis. The study conformed to the regulations of the Spanish Biomedical Research Act 14/2007, and data processing was conducted as per the EU Regulation 2016/679. 

The experts who participated in the study did so voluntarily and provided their consent prior to the first round. They received no financial compensation and were free to abandon the study at any time without the need for explanation. The questionnaire was completely anonymous in the sense that the experts consulted were unaware of the participation of the rest of the group. The Principal Investigator (PI) created an alphanumeric code in the first stage to identify the participants and protect their anonymity and affiliation. This information was only available to the PI of the project.

## 3. Results

### 3.1. Participant Characteristics by Round

All participants met the inclusion criteria, being clinical nurses with expertise in venous therapy, working in a second- or third-level hospital, and with extensive healthcare experience. The hospitals where the nurses were working were public hospitals and private state-contracted hospitals. It should also be noted that 50% of the participants were recruited through the Phlebitis Zero project, a programme which contemplates a package of measures related to patient safety. The main goal of the programme is to reduce PVC-related phlebitis and bacteriemia rates to the standard values established by international bodies, contributing to the minimization of adverse events. The programme is specifically aimed at hospital units, providing training for professional healthcare workers and facilitating the acquisition of a culture of safety.

A total of 47 people were initially invited to participate, with 30 participating in the first and second rounds, making a loss of 36.18%. In the third round, there were 27 participants, making a total loss of 42.56%. Table 1 shows the data of the final participants (Round 3) in detail. 

### 3.2. Results by Round

The results of the first round are presented in Table 2. From the analysis, it emerges that most of the participants assess phlebitis through symptomology/observation (25/30) and the use of the Maddox scale (23/30), with pain being the most prominent symptom (13/30). In relation to treatment, Burow’s solution leads the way (27/30), followed by catheter removal (10/30) and the use of pentosan polysulphate sodium ointment (Thrombocid^®^) (6/30). The most common response with respect to follow-up items was general monitoring (24/30).

In round 2, participants were asked whether they agreed with the response item (%) for each dimension (assessment, treatment, and follow-up) and to rate its importance (on a Likert scale from 0 to 4). The results are shown in Table 3, classified by dimension (assessment items, treatment items, and follow-up items). Among the assessment items, the Maddox scale stands out with an 80% agreement of experts, followed by the observation of symptoms (56.7%). In the second dimension, 70% of the experts agreed with catheter removal as a treatment option, 63.3% agreed with the application of Burow’s solution, and 30% with the application of pentosan polysulphate sodium ointment and cold compresses. In the follow-up dimension, 80% of the experts expressed agreement with general monitoring and 50% with monitoring and the assessment of symptoms. At slightly below 50%, the experts agreed (46.7%) that the insertion point should be monitored. Finally, it should be noted that some items disappear in this second round as they were not valued by the experts. These include size and some other items in the assessment dimension, warm compresses and simple cure in the treatment dimension, and doing nothing in the follow-up dimension. 

Finally, in the third round, the participants were asked to prioritise the importance of the different items included in the three dimensions explored. The results are presented in Table 4, ordered by CV. In the assessment dimension, six items (symptomatology/observation, reddening, the Maddox scale, induration, temperature, and pain) have a CV equal to or less than 30%, in the Treatment dimension, just two (catheter removal, pentosan polysulphate sodium ointment + application of cold), and in the follow-up dimension, only one (monitoring + temperature). This indicates a high degree of disparity in terms of expert opinion with respect to the other items in each dimension, which means that the scores assigned to each items vary considerably and are highly heterogeneous. In addition, the CV for each dimension was calculated, with treatment being the lowest with 29%, followed by assessment with 35%, and monitoring with 43%.

## 4. Discussion

The findings of this Spanish national Delphi study show the consensus and importance of activities related to the management of PVC-related phlebitis in the three dimensions explored of assessment, treatment, and follow-up. In general terms, the results identified a significant number of nursing actions in each of the dimensions. The only dimension with an acceptable CV was treatment (29%), meaning it was more homogeneous in terms of responses than the assessment dimension (35%) and the follow-up dimension (43%).

### 4.1. Assessment Dimension

In the present study, the expert participants highlighted direct observation as a measure of assessment (CV = 18). This concurs with studies such as that by Marsh et al. [28], who reported that nurses had high sensitivity in terms of identifying the complication and concluded that direct observation is a useful methodology for its diagnosis, or by Huang et al. [29], who found a high correlation between nurses’ visual assessment of phlebitis and ultrasound imaging.

The clinical profile of this complication is mainly characterised by pain, swelling, redness, induration, and palpable cord in the affected vein [8,30,31,32]. However, in practice, not all symptoms have been shown to have significant correlations. In the study by Mihala et al. [33], significant correlations were found between temperature and tenderness and swelling and erythema. In this study, only redness, induration, and pain had an acceptable CV (≤30%).

In relation to pain, experts highlighted its importance as a symptom and the need to treat it, which is in agreement with the literature [34,35,36]. Different phlebitis assessment systems, including the Maddox scale, the INS phlebitis scale, PVC Assess, or the VIP scale, include the symptom of pain [11]. Therefore, pain is an indispensable indicator of the possible PVC-related complications. Nurses can visually verify some signs, but other symptoms reported by the patient, such as irritation, may be considered less reliable and potentially related to a possible adverse effect of the administration of certain intravenous drugs [37,38] rather than PVC phlebitis.

The direct observation of signs and symptoms could potentially be problematic not only for assessment but also for treatment and follow-up if the items to be assessed are not specifically detailed. The use of scales would therefore be essential [7]. Several tools to assess the presence and severity of PVC-related phlebitis have been described in the literature, but guidelines are lacking, and further studies are recommended to obtain valid and reliable assessment tools [12]. In this study, the scale of preference of the experts was the Maddox scale, with 80% agreement and a CV of 28%. This tool first appeared in 1977 and was commonly used throughout the 1980s and 1990s, although it lacked psychometric properties. In 1998, it was adapted and renamed the VIP scale [11]. Therefore, it is somewhat surprising that the participants of the present study cited the Maddox scale, but not other clinically relevant scales [6].

### 4.2. Treatment Dimension

In terms of treatment, the item with the highest level of consensus (70%) and importance (CV = 0%) in this study was the removal of the PVC, which is in accordance with the guidelines published by the INS and in the scientific literature [39] as the first action to be taken. However, there is a lack of consensus on the subsequent treatment of symptoms.

Various therapeutic modalities available for PVC-related phlebitis have been investigated, both systemic and topical, with the latter being the most commonly used to control symptoms and treat patient discomfort [3]. Following the classification made by García-Expósito et al. [6], topical treatments can be differentiated into physical, phytotherapeutic, and pharmacological measures. According to the findings of the present study, the only physical measure used by professionals is the application of cold compresses in order to perform venous vasoconstriction and reduce oedema [40]. Some studies [30,41,42,43] concluded that the application of cold or hot compresses is favourable for the reduction of phlebitis, not because of the temperature effect but rather the humid environment achieved with them. The application of cold compresses together with pentosan polysulphate sodium ointment was the only treatment with a CV below 30% (26%). Pentosan polysulphate sodium ointment (Trombocid^®^ in Spain) in its presentation as an ointment (1 mg/g or 5 mg/g) is a product that belongs to the pharmacotherapeutic group of topical antivaricose products indicated for the local symptomatic relief of superficial venous disorders [44].

With respect to other products used for the treatment of phlebitis, although with low levels of agreement and CVs exceeding the pre-defined threshold of acceptability, Burow’s solution appeared as a treatment on its own (CV = 37%) or in combination with pentosan polysulphate sodium ointment (CV = 37%). Burow’s solution is a magistral pharmaceutical preparation (aluminium acetate), but the available scientific evidence on its efficacy is scarce [45,46]. Little evidence is also available on topical non-steroid anti-inflammatory agents as a pharmacological measure (CV = 49%). Although some authors [47] have reported a dual purpose of their use, namely, the relief of pain resulting from venous catheterisation and the prevention of phlebitis.

With respect to phytotherapeutic products and the non-reporting in this study of their use, this may be due to the country where the study was held [16], since in Spain, such products are not approved by the Agency for Medicines and Health Products (AEMPS as per its initials in Spanish). Similarly, the experts consulted in this study did not report the use of drugs such as organic heparinoids and heparins, which have been included in various studies [48,49,50,51,52]. 

### 4.3. Follow-Up Dimension

Follow-up actions were found to be highly non-specific. Monitoring plus temperature control was the only item with a CV below the 30% threshold (28%). Temperature was also reported in the assessment dimension. Other reported follow-up actions for PVC-related phlebitis include the monitoring of the insertion point (CV = 31%) and changing the PVC and dressing (CV = 36%). Monitoring the insertion point is an indispensable element for the control of phlebitis as this is where the first visible sign of phlebitis often appears [11,53], while the use of a dressing favours the better fixation of the catheter [53,54,55]. It should be noted that the INS guidelines published in 2021 [12] maintain the use of both transparent and gauze dressings, although the literature [16,56,57] highlights the preference of professionals and nursing students for transparent dressings. In addition, as in the findings of this study, other research [57,58,59] corroborates the importance of the daily monitoring of the vascular access device.

Follow-up actions such as elevating the limb (CV = 48%) to promote venous return, or monitoring + PVC change every 72 h (67%) were also reported. A routine PVC change every 72 h is not currently described as a recommended action to control phlebitis or other complications [39]. However, it is a practice that is performed in some hospital centres, as is routine dressing change [39].

Finally, two actions with a high heterogeneity of response were the request for insertion of a peripherally inserted central catheter (PICC) (CV = 62%) and the use of Doppler ultrasonography as a control (CV = 76%). The INS [12] recommends that, for phlebitis caused by the infusion of highly phlebogenous products, a PICC or another central vascular access device should be used. This disparity among the nursing experts can be attributed to the specific requirements for PICC insertion, with the need for a nursing team specifically trained in their insertion [60]. Limited information is available in the literature about the use of Doppler ultrasonography in this context. A study undertaken by Yabunaka et al. [61] identified the use of ultrasound for the assessment of subcutaneous tissue and the possible classification of the degree of subcutaneous oedema. 

### 4.4. Limitations of the Study

There are limitations inherent to this type of study as it is exploratory in nature and relies on the response of experts. For reasons expressed in other studies [22], the experts are both a limitation and a strength of a Delphi method and the interventions that are reported should serve as a starting point for other clinical intervention studies.

### 4.5. Implications for Clinical Practice and Future Research

The expert opinions that were gathered, together with the results of this study, clearly show the need to generate new evidence through rigorous studies in the field of vascular access and nursing care. However, such a need extends beyond research as it also implies the urgency of establishing homogeneous protocols and standardized diagnostic treatment approaches. The lack of uniformity in the current practices can lead to significant variability in patient care, potentially increasing health risks and compromising the quality of the care that is given.

Homogeneity in care staff training is another crucial consideration. Healthcare professionals require consistent and continually updated training to ensure they are equipped with the necessary knowledge and skills. Differences in training can lead to competency and clinical practice gaps, which can negatively impact patient outcomes.

Firstly, it is of fundamental importance to focus on the individual needs of each patient in order to improve the quality of the care that is given and to minimize any associated risks, thereby guaranteeing safe and personalized healthcare. In addition, the adequate documentation on future risk assessment is vital for proper follow-up and decision making, enabling preventative and proactive care that can substantially improve patient outcome. Secondly, there is a need for a critical review of the current practices. Healthcare professionals need to reflect on the implications of this and look for ways to improve and optimize the work they carry out in accordance with their experience and specialist knowledge.

Very few studies related to the management of PVC-related phlebitis have been published, and further research is needed in this field to increase the validity of the scientific evidence. The next step should be clinical studies. More evidence also needs to be generated in different contexts at the international level to promote a consensus among experts.

## 5. Conclusions

A significant number of nursing actions were reported by the participating experts in each of the three dimensions (assessment, treatment, and follow-up) considered in this Delphic study. The subsequent importance assigned to each of these actions by the experts varied widely. The importance of systematising a correct assessment of PVC-related phlebitis through the observation of symptoms (redness, induration, temperature, pain) and the use of assessment scales should be emphasised. The Maddox scale was the only phlebitis assessment scale highlighted by the experts.

Of the three dimensions considered in the present study, the treatment of PVC-related phlebitis had the greatest degree of homogeneity. The most notable measures were catheter removal and the use of cold compresses + pentosan polysulphate sodium ointment. Finally, with respect to follow-up actions, monitoring and temperature control was the only action with a CV below the established threshold of 30%. The other follow-up measures displayed a high degree of heterogeneity in the responses of the experts.

## Figures and Tables

**Figure 1 healthcare-12-00378-f001:**
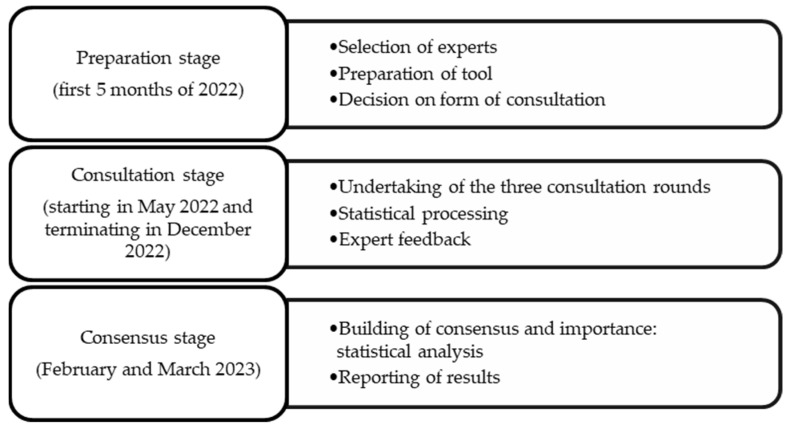
Procedure for carrying out the Delphi method.

**Table 1 healthcare-12-00378-t001:** Characteristics of the sample: number (n) and percentage (%) (n = 27).

Variables		n	%
Age *		45.0	11.3
Sex	Male	3	11.1
	Female	24	88.9
Years of experience *		23.0	11.6
Advanced training (professionals)	Speciality	8	29.6
	Master’s	13	48.1
	Postgraduate diploma	5	18.5
	None	1	3.7
Hospital level (place of work)	Second	10	37
	Third	17	63
Work experience (professionals)	Urology	1	3.7
	Infection Unit	4	14.8
	Paediatrics and mother and child healthcare	3	11.1
	Emergency Unit	4	14.8
	Internal Medicine	3	11.1
	Surgical Area	3	11.1
	Rotating shifts	2	7.4
	Intensive Care Unit	1	3.7
	Geriatrics	3	11.1
	Management	2	7.4
	Midwife	1	3.7
Autonomous Community	Catalonia	8	29.6
	Navarre	2	7.4
	Valencian Community	2	7.4
	Principality of Asturias	3	11.3
	Balearic Islands	1	3.7
	Cantabria	1	3.7
	Castilla and León	2	7.4
	Community of Madrid	2	7.4
	Region of Murcia	1	3.7
	Extremadura	1	3.7
	Galicia	2	7.4
	Aragón	1	3.7
	Canary Islands	1	3.7

* Mean and standard deviation (SD) in the variables of age and years of experience.

**Table 2 healthcare-12-00378-t002:** Round 1: responses and frequencies (n = 30).

Assessment Items	f *	Treatment Items	f	Follow-Up Items	f
Symptomology (observation)	25	Burow’s solution	27	General monitoring	24
Maddox Scale	23	Catheter removal	10	Monitoring + observation of symptoms	4
Pain	13	Pentosan polysulphate sodium ointment	6	None	4
Erythema	6	Burow’s solution + pentosan polysulphate sodium ointment	3	Monitoring + temperature control	2
Reddening	5	Pentosan polysulphate sodium ointment + application of cold	3	Monitoring of insertion point	2
Swelling	5	Cold compresses	2	Monitoring + PVC * change every 72 h	2
Oedema	3	Warm compresses	1	Monitoring + pain control	2
Venous cord	3	Non-steroid anti-inflammatories (ointment)	1	Monitoring + Doppler ultrasonography	1
Induration	3	Non-steroid anti-inflammatories (oral)	1	Monitoring + other measures	1
Warmth	2	Simple cure	1	Change PVC and dressing	1
Irritation	2			Elevating the limb	1
Size	1			Request PICC * insertion	1
Temperature	1				
Other	1				
Scale + symptomology	1				

* f = frequency; PVC = peripheral venous catheter; and PICC = peripherally inserted central catheter.

**Table 3 healthcare-12-00378-t003:** Round 2: importance (M and SD) and agreement (%) (n = 30).

	Importance			Agree		Importance			Agree		Importance			Agree
Assessment Items	M *	SD *	Mdn *	(%)	Treatment Items	M	SD	Mdn	(%)	Follow-Up Items	M	SD	Mdn	(%)
Maddox scale	3.4	1.2	4	80	Catheter removal	3.8	0.64	4	70	General monitoring	3.7	0.81	4	80
Symptomology (observation)	3.9	0.43	4	56.7	Burow’s solution	2.6	1.52	3	63.3	Monitoring + observation of symptoms	3.6	0.67	4	50
Pain	3.3	0.92	4	43.3	Pentosan polysulphate sodium ointment + application of cold	2.7	1.39	3	30	Monitoring of insertion point	3.8	0.54	4	46.7
Erythema	3.4	0.88	4	33.3	Cold compresses	2.5	1.15	3	20	Change PVC * and dressing	2.4	1.43	2	16.7
Venous cord	3.3	1.37	4	30	Pentosan polysulphate sodium ointment	1.8	1.49	1.5	13.3	Monitoring + PVC change every 72 h	1.8	1.62	1	13.3
Reddening	2.9	1.25	3	30	Burow’s solution + Pentosan polysulphate sodium ointment	2.2	1.49	2.5	10	Monitoring + pain control	3.2	0.96	3.5	10
Induration	3.2	1.35	4	26.7	Non-steroid anti-inflammatories (ointment)	2.1	1.35	2	6.7	Monitoring + temperature control	3	1.14	3	10
Temperature	2.9	1.21	3	23.3	Non-steroid anti-inflammatories (oral)	0.6	0.88	1	3.3	Elevating the limb	1.7	1.55	1.5	10
Swelling	2.8	1.32	3	16.7						Request PICC * insertion	1.2	1.26	1	10
Oedema	2.7	1.37	3	16.7						Monitoring + Doppler ultrasonography	1.1	0.96	1	6.7
Colour	2.3	1.31	2.5	16.7						Monitoring + other measures	0.9	1.24	0	6.7
Scale + symptomology	2.2	1.87	3	13.3										
Irritation	1.7	1.12	1.5	3.3										

* M = mean; SD = standard deviation; Mdn = median; PVC = peripheral venous catheter; and PICC = peripherally inserted central catheter.

**Table 4 healthcare-12-00378-t004:** Round 3: prioritisation of importance (M, SD, and CV) (n = 27).

Assessment Item	M *	SD *	CV * (%)	Mdn *	Treatment Item	M	SD	CV (%)	Mdn	Follow-Up Item	M	SD	CV (%)	Mdn
Symptomology (observation)	11.1	2.1	18	12	Catheter removal	8.0	0.0	0	8	Monitoring + temperature control	5.9	1.7	28	6
Reddening	7.3	2.0	27	8	Pentosan polysulphate sodium ointment + application of cold	5.2	1.4	26	4.5	Monitoring of insertion point	7.9	2.5	31	9
Maddox scale	11.2	3.2	28	12	Cold compresses	5.0	1.6	32	6	Monitoring + observation of symptoms	8.3	2.8	33	9.5
Induration	7.7	2.2	28	7	Pentosan polysulphate sodium ointment	3.65	1.3	35	4	Monitoring + pain control	6.2	2.2	35	6.5
Temperature	6.0	1.8	30	6	Burow’s solution	5.6	2.1	37	7	Change PVC * and dressing	6.6	2.4	36	3
Pain	9.3	2.8	30	10	Burow’s solution + Pentosan polysulphate sodium ointment	4.0	1.5	37	3.5	General monitoring	7.8	3.5	44	9
Venous cord	8.7	2.7	31	9	Non-steroid anti-inflammatories (ointment)	2.65	1.3	49	2	Elevating the limb	4.5	2.2	48	4
Erythema	7	2.5	35	7	Non-steroid anti-inflammatories (oral)	1.8	1.4	77	2	Request PICC * insertion	3.7	2.3	62	3
Oedema	4.8	2.2	45	6						Monitoring + other measures	2.6	1.7	65	2
Colour	4.2	2.2	52	4						Monitoring + PVC change every 72 h	4.9	3.3	67	5
Scale + symptomology	8.1	4.6	56	10						Monitoring + Doppler ultrasonography	3.2	2.45	76	2
Swelling	4.3	2.5	58	5										
Irritation	1.6	1.6	100	1										

* M = mean; SD = standard deviation; CV = coefficient of variation; Mdn = median; PVC = peripheral venous catheter; and PICC = peripherally inserted central catheter.

## Data Availability

The data analysed during the current study are not publicly available due to privacy restrictions but are available from the corresponding authors on reasonable request.

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
