# Peer review of "Assessment, Treatment, and Follow-Up of Phlebitis Related to Peripheral Venous Catheterisation: A Delphi Study in Spain"

_healthcare, 2024, doi:10.3390/healthcare12030378_

Round 1
Reviewer 1 Report
Comments and Suggestions for Authors
Journal: Healthcare (ISSN 2227-9032)
Manuscript ID: healthcare-2804232
Type: Article
Title: Agreement on actions to be taken and their respective importance in the assessment, treatment and follow-up of phlebitis secondary to peripheral venous catheterisation: a Delphi study at national level in Spain
The article has been examined and the following suggestions have been made.
1. Article Title
The article title should contain short and understandable expressions that summarize the subject. The article title should be rearranged.
2. In the the Abstract Section;
1. Before the term 'epidemiological perspective' in the first sentence in the Background section, explain this term with an explanatory sentence. The sentence containing the term has no relevance to 'epidemiological perspective'.
2. The technique expressed as ': Three-round Delphi technique' in the Method section should be corrected by first explaining the technique and then naming it. In addition, there is no information on how many people were included in the study, how the objectivity of the test was ensured (such as the professional training of the participants, whether they had conducted a similar survey before, the participant's experience, etc.) and the inclusion criteria. The article should be corrected in this respect.
3-In the results section, the parameters that are meaningful in the questionnaire should be explained. The article should be corrected in this respect.
3.In the Introduction Section
First, state the primary and secondary purposes of your study and include a broader section explaining the reason for using your questionnaire. Summarize other consensuses referenced in this section and shorten the section.
4. In the Materials and Methods Section
In the '2.1 Design' section, explain the term 'Delphi technique' in terms of the population in which it is used, the results expected from this technique, and its validity and reliability.
2. In the material method section, indicate the qualitative and quantitative evaluation (inclusion/exclusion) of the participants to whom the test was applied.
3- Specify the cut-off value for each value to be measured in the Questionnaire, or if there is a cut-off value, include a section explaining its correlation with the study.
5. In the Results Section
1. Explain the term 'Phlebitis Zero project' used in this section.
6. A title 'Statistical Method' should be created in the article. All methods used (including cut-off point of parameters, reliability analysis, Power Analysis) should be included in this section.
7. In the Discussion Section
1- In this section, new headings have been created and many of the same expressions have been included. Headings should be removed and the results obtained should be synthesized with current literature information without falling into repetition.
8. In the Conclusion Section
1-In this section, existing word errors must be corrected and the article must be reviewed in this respect. If the term 'Maddox scale' is a method used in the study, it should be explained in the 'Material-Method' section and an explanation should be made regarding its relevance to the study.
Author Response
Reviewer 1
Journal: Healthcare (ISSN 2227-9032)
Manuscript ID: healthcare-2804232
Type: Article
Title: Agreement on actions to be taken and their respective importance in the assessment, treatment and follow-up of phlebitis secondary to peripheral venous catheterisation: a Delphi study at national level in Spain
The article has been examined and the following suggestions have been made.
We would like to express our gratitude for all the comments. We consider that they have helped enormously in improving our paper.
- Article Title
The article title should contain short and understandable expressions that summarize the subject. The article title should be rearranged.
Following this suggestion, we propose a shorter and easier to understand title.
Title: Assessment, treatment and follow-up of phlebitis related to peripheral venous catheterisation: a Delphi study in Spain
- In the the Abstract Section;
- Before the term 'epidemiological perspective' in the first sentence in the Background section, explain this term with an explanatory sentence. The sentence containing the term has no relevance to 'epidemiological perspective'.
We have rewritten the Background section, focusing more on the patient and phlebitis.
Page 1, line 20-3
- The technique expressed as ': Three-round Delphi technique' in the Method section should be corrected by first explaining the technique and then naming it. In addition, there is no information on how many people were included in the study, how the objectivity of the test was ensured (such as the professional training of the participants, whether they had conducted a similar survey before, the participant's experience, etc.) and the inclusion criteria. The article should be corrected in this respect.
We have now given some brief information on the participants but are limited in the Abstract by the number of words suggested by the Journal.
Page 1, line 25-6
3-In the results section, the parameters that are meaningful in the questionnaire should be explained. The article should be corrected in this respect.
Thank you for this helpful comment. We have now expanded this section to include the different aspects highlighted by the participants.
Page 1, line 31-5
3.In the Introduction Section
First, state the primary and secondary purposes of your study and include a broader section explaining the reason for using your questionnaire. Summarize other consensuses referenced in this section and shorten the section.
We have reorganized the Introduction section and removed some information. We consider that this reorganization helps the reader better follow the elements that are subsequently explored on the management of phlebitis in the three dimensions considered (assessment, treatment and follow-up. These three dimensions are presented as secondary objectives of the study.
Page 1-2, line 42-94
- In the Materials and Methods Section
In the '2.1 Design' section, explain the term 'Delphi technique' in terms of the population in which it is used, the results expected from this technique, and its validity and reliability.
Many thanks for this comment. We have revised the information on the Delphi method. A Delphi study is classified initially as exploratory in type as it is based on expert opinion and its level of evidence is therefore limited in comparison to other types of study. It is also classified as a qualitative study and is not measured in quantitative criteria.
Following your suggestion, we have introduced new references which explain this methodology and have included more details on the methodological quality of the study. We would also like to highlight the fact that one of the members of our research team, Dr. Reguant, is an expert in the Delphi technique, Reguant Alvarez and Torrado Fonseca (2016).
Page 3, line 100-1
Page 5, line 205-9
Reguant Alvarez, M.; Torrado Fonseca, M. El Método Delphi. REIRE Rev. Innovació Recer. En Educ. 2016, 87–102, doi:10.1344/reire2016.9.1916.
- In the material method section, indicate the qualitative and quantitative evaluation (inclusion/exclusion) of the participants to whom the test was applied.
We have revised the inclusion criteria and added the exclusion criteria.
Page 3, line 125-6
Specify the cut-off value for each value to be measured in the Questionnaire, or if there is a cut-off value, include a section explaining its correlation with the study.
Our apologies, but we are unsure if we have correctly understood this comment. The cut-off values appear in section 2.6. We have revised the text to verify that the percentages and CV are correct. The values are based on previously used values in the literature.
- In the Results Section
- Explain the term 'Phlebitis Zero project' used in this section.
We have explained in greater detail the Phlebitis Zero project.
Page 5, line 229-34
- A title 'Statistical Method' should be created in the article. All methods used (including cut-off point of parameters, reliability analysis, Power Analysis) should be included in this section.
The information requested can be found in section 2.6, although it should be noted the criteria differ somewhat from quantitative studies as the methodology is qualitative in nature. It is therefore presented as a social research technique which is qualitative in nature in accordance with the opinion of experts such as Landeta and Lertxundi (2023) and Niederberger and Spranger (2020)
Nonetheless, following this proposal of the reviewer, we have tried to add further details to better understand this potentially problematic aspect.
Page 4, line 180-2
Landeta, J., & Lertxundi, A. (2023). Quality indicators for Delphi studies. Futures & Foresight Science, e172
Niederberger, M., & Spranger, J. (2020). Delphi technique in health sciences: A map. Frontiers in Public Health, 8, 457.
- In the Discussion Section
- In this section, new headings have been created and many of the same expressions have been included. Headings should be removed and the results obtained should be synthesized with current literature information without falling into repetition.
We have revised this section and simplified the information it contains as there was indeed excessive repetition as the reviewer correctly pointed out.
However, we would request that the reviewer allow us to maintain the different sections as we consider they are of help to the reader in their general understanding. The reviewer, however, will see that we have eliminated some parts and rewritten others.
Page 9-12, line 293-451
- In the Conclusion Section
1-In this section, existing word errors must be corrected and the article must be reviewed in this respect. If the term 'Maddox scale' is a method used in the study, it should be explained in the 'Material-Method' section and an explanation should be made regarding its relevance to the study.
We have rewritten the section to avoid any possible confusions. The Maddox scale is a phlebitis assessment scale.
Page 12, line 458-9
We would like to thank the reviewer once again for the time and effort put into the review of our manuscript. It has been a pleasure to receive such expert comments. If any further clarifications are required, please let us know.
Reviewer 2 Report
Comments and Suggestions for Authors
Dear Authors! My congratulations for the interesting study you conducted. I have just some minor remarks.
1. I'd prefer you mention in the abstract main all six items obtained after third round. This would be useful for readers.
2. I'd prefer you to shorten Introduction by removing not crucial details.
3. I'd prefer you to shorten conclusions by mentioning only your own findings.
Other remarks are:
Line 55. Patients have phlebitis, not authors.
Lines 205-207. These lines seem to be not a paper part.
Table 1. What does percentage mean for age and years of experience?
Table 1. Asterisk for age denotes “mean and standard deviation”? But, where is the SD in the table?
Comments on the Quality of English LanguageI'm not a native English speaker myself, but it seems to me that the text was written not by native also. I believe it would be useful to ask a native speaker to check the text.
Author Response
Reviewer 2
Journal: Healthcare (ISSN 2227-9032)
Manuscript ID: healthcare-2804232
Dear Authors! My congratulations for the interesting study you conducted. I have just some minor remarks.
Many thanks for your kind words of encouragement and the expert comments that you have made and which, we consider, have contributed greatly to improving our manuscript.
- I'd prefer you mention in the abstract main all six items obtained after third round. This would be useful for readers.
Thank you for this comment which we appreciate helps the reader considerably. We have included the different aspects highlighted by the participating experts.
Page 1, line 31-5
- I'd prefer you to shorten Introduction by removing not crucial details.
Following this suggestion, we have rewritten parts of the Introduction, eliminating some text and separating some elements into paragraphs to allow a better understanding of them.
Page 1-2, line 42-94
- I'd prefer you to shorten conclusions by mentioning only your own findings.
We have rewritten the Conclusions section following your suggestion and have made them more specific and briefer.
Page 12, line 453-66
Other remarks are:
- Line 55. Patients have phlebitis, not authors.
We have rewritten this part of the text, which was indeed confusing.
Page 2, line 57
- Lines 205-207. These lines seem to be not a paper part.
Please accept our apologies for this oversight. These lines were mistakenly left in the text. They have now been removed.
- Table 1. What does percentage mean for age and years of experience? Table 1. Asterisk for age denotes “mean and standard deviation”? But, where is the SD in the table?
The majority of the items in Table 1 are recorded in terms of frequency and percentage. The two items marked with an asterisk (Age and Years of experience) indicate that the values are recorded in terms of the mean and standard deviation. We have now indicated this more precisely in the table.
Page 6. Table 1, line 244
- Comments on the Quality of English Language
I'm not a native English speaker myself, but it seems to me that the text was written not by native also. I believe it would be useful to ask a native speaker to check the text.
As we are also not native speakers of English, we have contracted the services of a professional with over 20 years of experience in the correction and editing of academic scientific texts to ensure that there no mistakes in the grammar or language used. The manuscript has now been fully checked and revised for its English content.
We would like to thank the reviewer again for the time and effort put into the review of our manuscript. Please do not hesitate to let us know if any further clarifications are required.

Reviewer 3 Report
Comments and Suggestions for Authors
Dear authors.
It has been a real pleasure to have had the opportunity to review your manuscript.
In the attached document (PDF) I have generated comments with the aim of providing feedback and making the appropriate modifications to the text (please mark in a different colour to identify the changes).
The biggest doubt I have is related to ethical aspects of the work and the response of the CEIm (Spain).
Best regards.
----
LINE 123-125: indicates declined participation 40% ( 40% of a population of 47 subjects = 18.8, hence participation 28.2 subjects). check percentage, subjects cannot have decimals. See line 112-113.
LINE 127: Please indicate which tool you use to disseminate the online survey (Google Forms, SurveyMonkey, .....).
LINE 129: please indicate reason for exclusion.
LINE 110: In the methodology it is the first time you refer to the hospitals selected for study please specify in this first time which types of hospitals (public, private, both...).
LINE 161: What type of parameter did you use for the calculation of the concordance percentage?
LINE 212-214. review together with line 112-113.
LINE 194-203 (ethical considerations): I have some doubts, for this reason, I ask you to provide a response to this comment and to add in the manuscript the data in reference to committee authorisation or the reason why it is not requested.
1. Are the contact details of the respondents (email) in the open (PhlebitisZero)?
2. In these emails are institutions involved (emails of nurses in your institution - xxx@hospitalxxx.com) or are all emails personal?
3. If the nurses belong to institutions, was authorisation requested from the Ethics Committee of reference?
4. Another possibility, if the study is created from the university (1st author), was authorisation requested from the Human Research Ethics Committee (for research on human beings, with biological samples of human origin and with personal data)?
LINE 455-463: Intitutional Review Board Statement. In this section, it indicates that the committee did not consider it necessary to request authorisation.
*Due to all the points mentioned above, and the possible involvement of professionals from institutions, data protection and consent to participate, I have doubts, which is why I leave this point to the editor of the journal's assessment.
REFERENCES: It is recommended to read the reference style guide "Healthcare". ACS. Check all references. E.G.
B. Almirante, ‘Diagnóstico y tratamiento de las bacteriemias asociadas con el uso de los catéteres vasculares: que 473 aporta una nueva guía de práctica clínica’, Med. Intensiva, vol. 42, no. 1, pp. 1–4, Jan. 2018, doi: 10.1016/j.me-474 din.2017.12.009.
Almirante, B. Diagnóstico y tratamiento de las bacteriemias asociadas con el uso de los catéteres vasculares: que aporta una nueva guía de práctica clínica, Med. Intensiva, 2018, 42, 1. doi: 10.1016/j.me-474 din.2017.12.009.

Author Response
Reviewer 3
Journal: Healthcare (ISSN 2227-9032)
Manuscript ID: healthcare-2804232
Dear authors.
It has been a real pleasure to have had the opportunity to review your manuscript.
In the attached document (PDF) I have generated comments with the aim of providing feedback and making the appropriate modifications to the text (please mark in a different colour to identify the changes).
The biggest doubt I have is related to ethical aspects of the work and the response of the CEIm (Spain).
Best regards.
We would like to thank the reviewer for these words and the comments and suggestions made. We consider they have helped to considerably improve our manuscript.
With respect to the ethical aspects, below we provide some details about the committee in response to the doubts that have been raised, demonstrating that the ethical aspects of the work are ensured through observance of the prevailing legislation.
In Spain, a Drug Research Ethical Committee (CEIm by its initials in Spanish) is not an internal committee of a hospital centre but rather the highest assessment body of a research study. Consequently, its rulings are definitive in relation to the aspects it is evaluating. The CEIm of the Arnau de Vilanova University Hospital of the Lleida Territorial Health Service Management (https://www.gss.cat/ca/organitzacio/direccio/territorial/recerca/ceim) is an authorised centre, in accordance with Royal Decree 1090/2015, for the issuing of rulings on clinical research at national level. It is therefore regulated by the Government of Spain https://www.aemps.gob.es/medicamentos-de-uso-humano/investigacion_medicamentos/investigacionclinica_ceim/?lang=gl.
We would also like to comment that the ruling of the CEIm is in line with those issued for other Delphi studies, in that they tend to be of a low level of interventionism and may often not require informed consent. Delphi studies involve the voluntary participation of experts and confidential patient data are not required.
Steel, M., Seaton, P., Christie, D., Dallas, J., & Absalom, I. (2021). Nurse perspectives of nurse-sensitive indicators for positive patient outcomes: A Delphi study. Collegian, 28(2), 145–156. https://doi.org/10.1016/j.colegn.2020.02.009
Koch D, Kutz A, Conca A, Wenke J, Schuetz P, Mueller B. The relevance, feasibility and benchmarking of nursing quality indicators: A Delphi study. J Adv Nurs. 2020 Dec;76(12):3483-3494. doi: 10.1111/jan.14560.
We have now included some additional information on the ruling of the CEIm to avoid any possible confusion.
Page 13, Line 481-2
LINE 123-125: indicates declined participation 40% ( 40% of a population of 47 subjects = 18.8, hence participation 28.2 subjects). check percentage, subjects cannot have decimals. See line 112-113.
Our apologies for this oversight on our part. The reviewer is of course completely correct. We have now rewritten the text where the error was and have checked the % values.
Page 3, line 133-35
LINE 127: Please indicate which tool you use to disseminate the online survey (Google Forms, SurveyMonkey, .....).
The Research Electronic Data Capture (REDCap) web platform was used. This information is reported on line 153, and we have now included the address of the corresponding website.
LINE 129: please indicate reason for exclusion.
Thank you for this comment, we have now included the exclusion criteria.
Page 3, Line 125-6
LINE 110: In the methodology it is the first time you refer to the hospitals selected for study please specify in this first time which types of hospitals (public, private, both...).
We have now included this information on the type of hospital.
Page 3, Line 119-20
Page 5, Line 227-8
LINE 161: What type of parameter did you use for the calculation of the concordance percentage?
The information requested can be found in section 2.6. We would like to thank the reviewer for this comment as it enabled us to detect an error in the original manuscript in that the heading of section 2.5 was incorrect. Section 2.5 only concerns the data collection and how the information was presented to the experts in the different rounds. Details about the analysis of the data and the parameter used to calculate the consensus are in section 2.6.
Page 4, Line 151,180-2
LINE 212-214. review together with line 112-113.
As suggested by the reviewer, we have checked the lines as requested and added some information.
LINE 194-203 (ethical considerations): I have some doubts, for this reason, I ask you to provide a response to this comment and to add in the manuscript the data in reference to committee authorisation or the reason why it is not requested.
We would like to make the following comments with respect to the doubts that have been raised:
The study presented in the manuscript forms part of a doctoral thesis registered in the Healthcare Programme of the Doctoral School of the University of Lleida (UdL). The UdL is registered in the CEIm of the Arnau de Vilanova University Hospital as a non-healthcare centre (https://www.gss.cat/sites/gss.cat/files/Centres-adscrits-i-a-mbit-d-actuacio-del-CEIm.pdf). The study did not use any data related to the UdL.
A specific email was created for the study. The institution’s email was not used. As an element of good practice, the data and its processing were securely stored and could only be accessed by the PI. As professors in the UdL, the authors were obliged to follow the guidelines established in the Good Practices in Research and Publication drawn up by the UdL (https://www.recercaitransferencia.udl.cat/export/sites/Recerca/ca/.galleries/docs/Buenas-Practicas-y-RRI-UdL_def.pdf).
The participants gave only general data, which are not directly specified. Therefore, their qualification as an expert in the study under investigation is not directly related to their current occupation but rather their professional experience in the field. The institutions where they work are not named and are impossible to identify, as is also the case for the participants.
As described in the manuscript, contact was made based on publicly available information. Contact was also made with the Phlebitis Zero organization (https://resistenciaantibioticos.es/sites/default/files/documentos/programa_flebitis_zero.pdf) to inform them of our study and to invite their members to participate. Other participants were invited through the email (corresponding author) of publications or other public projects. All the experts were informed that their participation was voluntary in nature, that they would receive no financial or other type of compensation, and that they could withdraw from the study at any time without the need for any explanation.
- Are the contact details of the respondents (email) in the open (PhlebitisZero)?
- In these emails are institutions involved (emails of nurses in your institution - xxx@hospitalxxx.com) or are all emails personal?
- If the nurses belong to institutions, was authorisation requested from the Ethics Committee of reference?
- Another possibility, if the study is created from the university (1st author), was authorisation requested from the Human Research Ethics Committee (for research on human beings, with biological samples of human origin and with personal data)?
LINE 455-463: Intitutional Review Board Statement. In this section, it indicates that the committee did not consider it necessary to request authorisation.
*Due to all the points mentioned above, and the possible involvement of professionals from institutions, data protection and consent to participate, I have doubts, which is why I leave this point to the editor of the journal's assessment.
We have now included complementary information to resolve these doubts in the new version of the manuscript.
Page 3, Line 129-34
Page 7, Line 212-3
Page 7, Line 217-8
Page 7, Line 221
REFERENCES: It is recommended to read the reference style guide "Healthcare". ACS. Check all references. E.G.
- Almirante, ‘Diagnóstico y tratamiento de las bacteriemias asociadas con el uso de los catéteres vasculares: que 473 aporta una nueva guía de práctica clínica’, Med. Intensiva, vol. 42, no. 1, pp. 1–4, Jan. 2018, doi: 10.1016/j.me-474 din.2017.12.009.
Almirante, B. Diagnóstico y tratamiento de las bacteriemias asociadas con el uso de los catéteres vasculares: que aporta una nueva guía de práctica clínica, Med. Intensiva, 2018, 42, 1. doi: 10.1016/j.me-474 din.2017.12.009.
The reviewer is perfectly correct. We applied a variant of the Vancouver style. We have now checked and adapted all the references.
We would like to thank the reviewer once again for the time and effort put into the review of our manuscript. If any further clarifications are required, please let us know.

Round 2
Reviewer 1 Report
Comments and Suggestions for Authors
Journal: Healthcare (ISSN 2227-9032)
Manuscript ID: healthcare-2804232
Type: Article
Title: Agreement on actions to be taken and their respective importance in the assessment, treatment and follow-up of phlebitis secondary to peripheral venous catheterisation: a Delphi study at national level in Spain
The article has been examined and the following suggestions have been made.
1-It has been observed that the 'Statistical Analysis' section has been removed from the article. Before the 'Results' section of the article, the statistics section should be added by creating a separate heading, as stated in the previous revision proposal.
2- If the 'Madox scale' mentioned in the sentence 'The Maddox scale was…' in the 'Conclusion' section was used in the evaluation of the findings in the study practice, it should be included in the material method section. The relevant correction suggestion was also reported in the previous revision note.
Author Response
Journal: Healthcare (ISSN 2227-9032)
Manuscript ID: healthcare-2804232
R2
Type: Article
Title: Agreement on actions to be taken and their respective importance in the assessment, treatment and follow-up of phlebitis secondary to peripheral venous catheterisation: a Delphi study at national level in Spain
The article has been examined and the following suggestions have been made.
We have revised the aspects you mention. We have probably not explained ourselves well.
1-It has been observed that the 'Statistical Analysis' section has been removed from the article. Before the 'Results' section of the article, the statistics section should be added by creating a separate heading, as stated in the previous revision proposal.
We have never had the Statistical Analysis section before as we had it included in section 2.6 as Consensus Stage. We will use the title Statistical Analysis following your suggestion and we also add it to figure 1. Thank you for your contribution.
2- If the 'Madox scale' mentioned in the sentence 'The Maddox scale was…' in the 'Conclusion' section was used in the evaluation of the findings in the study practice, it should be included in the material method section. The relevant correction suggestion was also reported in the previous revision note.
We feel we haven’t explained ourselves well. The Maddox scale emerges from the results (for this reason, we did not put in the method section it). The expert nurses contribute it as a scale of assessment of phlebitis in the first dimension (Table 2,3,4).
There are various scales for the assessment of phlebitis, but the experts only mention that one. That’s why it appears, and it is results are highlight in Conclusions.
Thanks again for your time and we remain at your disposal.
